# Solving the Time- and Frequency-Multiplexed Problem of Constrained Radiofrequency Induced Hyperthermia

**DOI:** 10.3390/cancers12051072

**Published:** 2020-04-25

**Authors:** Andre Kuehne, Eva Oberacker, Helmar Waiczies, Thoralf Niendorf

**Affiliations:** 1MRI.TOOLS GmbH, Robert-Roessle-Str. 10, 13125 Berlin, Germany; 2Berlin Ultrahigh Field Facility (B.U.F.F.), Max-Delbrück-Center for Molecular Medicine in the Helmholtz Association, Robert-Roessle-Str. 10, 13125 Berlin, Germany; 3Clinic for Radiation Oncology, Charité Universitätsmedizin, 13125 Berlin, Germany

**Keywords:** RF hyperthermia, thermal intervention, field shaping, field focusing, RF applicator, hyperthermia treatment planning, thermal magnetic resonance, semidefinite programming

## Abstract

Targeted radiofrequency (RF) heating induced hyperthermia has a wide range of applications, ranging from adjunct anti-cancer treatment to localized release of drugs. Focal RF heating is usually approached using time-consuming nonconvex optimization procedures or approximations, which significantly hampers its application. To address this limitation, this work presents an algorithm that recasts the problem as a semidefinite program and quickly solves it to global optimality, even for very large (human voxel) models. The target region and a desired RF power deposition pattern as well as constraints can be freely defined on a voxel level, and the optimum application RF frequencies and time-multiplexed RF excitations are automatically determined. 2D and 3D example applications conducted for test objects containing pure water (r_target_ = 19 mm, frequency range: 500–2000 MHz) and for human brain models including brain tumors of various size (r_1_ = 20 mm, r_2_ = 30 mm, frequency range 100–1000 MHz) and locations (center, off-center, disjoint) demonstrate the applicability and capabilities of the proposed approach. Due to its high performance, the algorithm can solve typical clinical problems in a few seconds, making the presented approach ideally suited for interactive hyperthermia treatment planning, thermal dose and safety management, and the design, rapid evaluation, and comparison of RF applicator configurations.

## 1. Introduction

Vigorous fundamental and (bio)engineering research into electromagnetic field radiation induced heating of tissue has culminated in an enormous body of literature [1,2,3,4]. Localized tissue heating induced by electromagnetic fields (EMF) has a wide array of applications ranging from thermal therapy as a potent sensitizer of chemo- and radiotherapy to the controlled release of therapeutics from a nano-carrier [5,6,7,8,9,10,11,12]. Radiofrequency (RF) heating relies on the interference of electric fields produced by multiple independent RF sources, which are sought to constructively interfere in the targeted heating volume while keeping RF power deposition outside the target to a minimum to preserve healthy tissue. This requirement has spurred consideration of the physics and EMF simulations and has motivated the development of RF applicator concepts to advance thermal intervention with the focal point quality being governed by the radiation pattern of the single RF transmit element, the RF channel count, and the thermal intervention radiofrequency of the RF applicator [13,14,15,16].

The problem of finding the appropriate excitations for each individual RF channel in an RF applicator to generate the desired specific absorption rate (SAR) pattern is nonconvex. Finding the global optimum has proven to be challenging and computationally intensive. Time reversal techniques can be used to maximize RF power deposition at one point, but cannot constrain SAR in the remaining regions [17,18]. Another approach is to forego direct constraints on localized RF power dissipation in healthy tissues and instead try to indirectly limit unwanted SAR hot-spots outside the target region via the tumor-to-hotspot-SAR-ratio [19]. An analytical solution exists to maximize average RF power inside a target region relative to the total absorbed RF power (SAR amplification factor, SAF). However, this approach does not preclude strong unwanted localized RF heating effects [13,20,21]. Closely related approaches try to indirectly suppress unwanted hotspots by iteratively reweighting the integrated power absorbed in healthy tissues and solving a generalized eigenvalue problem [21,22,23]. Investigations into a time-multiplexed application of a set of interference patterns [23,24,25] and broadband multi-frequency applicators [26,27,28,29,30] have found a potential use for these extensions. However, a rigorous algorithm for selecting the globally optimal frequencies and/or multiplexed excitations does not exist to the knowledge of the authors. A number of proposed methods seek to apply array synthesis techniques [31] to directly shape the electric fields in a desirable pattern. The alternating projections algorithm seeks to find the optimum fit of a feasible field pattern to a desired target, albeit without guarantee of a global optimum and precluding a direct constraint on undesired hotspots [32,33]. Existing convex approaches to the field shaping problem linearize the problem by using only the strongest field component at a single or very few point(s) [34,35,36,37], which will not lead to the optimum solution if the assumption of a single dominating component does not hold true in the target region.

A robust globally optimal solution to the electric field focusing problem would not only significantly benefit the treatment planning and the treatment efficacy of targeted RF heating induced hyperthermia, but would also enhance the design and evaluation process of RF applicators. Different RF applicator designs could be readily compared, and their relative performance accurately benchmarked without the added uncertainty of a non-optimal solution.

Recognizing this opportunity, this work rigorously derived a convex formulation of the time- and frequency-multiplexed constrained RF heating problem without unwarranted simplifications and presents an iterative algorithm to efficiently find its globally optimum solution. Since our proposed approach provides full control over not only single components, but the whole vector field distribution using time- and frequency-multiplexing, we termed it multiplexed vector field shaping (MVFS). The feasibility of this approach was demonstrated in electromagnetic field (EMF) simulations using a 2-dimensional circular water phantom irradiated by 32 broadband localized RF sources created by a plane wave incident on a small aperture. To advance from phantom setups to a clinical setup, 3D evaluation of the proposed algorithm was performed in a human brain model including spherical tumors of various sizes and locations as a target volume using a synthetic broadband RF applicator comprising a 40-channel helmet grid design.

## 2. Materials and Methods

### 2.1. Problem Statement

For time-harmonic electromagnetic fields generated by an array of *N* interfering sources, power deposition *P* and related metrics such as the specific absorption rate (SAR) can be expressed using quadratic forms [20,38]:(1)$$P\;=\;x^{H}Qx$$

Here, *x* denotes the complex (amplitude and phase) excitation vector used to drive the array, superscript *H* represents the Hermitian transpose, and ***Q*** is a *N × N* positive-semidefinite (psd) power correlation matrix obtained by forming correlation integrals of the electric fields inside lossy tissues. The elements *q_ij_* of the matrix ***Q*** are hence calculated via (2)$$q_{ij}\;=\;\frac{1}{2}\overset{}{\underset{V}{\int \int \int }}\sigma \left(r\right)E_{i}^{*}\left(r\right)E_{j}\left(r\right)dV$$
where indices *i*, *j* denote the source number; *σ*(***r***) the electrical conductivity; and ***E_i_***(***r***) represents the electrical field of the i-th source using a unit excitation. By choosing the volume of integration, matrices representing said volume can be formed such as a tumor volume or cubes containing a given amount of tissue mass (e.g., 1 g or 10 g) in order to calculate spatially averaged local SAR.

The goal in localized RF heating applications is to deposit enough RF power in the desired target region in order to induce localized heating, while keeping RF power deposition in the remaining regions below a certain threshold to avoid unwanted tissue damage. Due to the additional complexities of accurate temperature modeling for in-vivo scenarios, local RF power deposition (i.e., local SAR) is typically used as a proxy for heating. Simple temperature rise models based on the linear bioheat transfer equation [39] without any time- or temperature-dependent properties can be formulated using a similar matrix formalism [23,40,41], thus allowing our approach to be directly applied to constrained temperature optimization. Here, we focused on SAR or RF power.

As both the objective function (SAR in target region) and the constraints (SAR outside target region) are represented by quadratic forms, the optimization problem to be solved falls within the class of quadratically constrained quadratic programs (QCQP). With the objective matrix ***Q***, constraint matrices ***S**_i_*, and associated constraint limits *c_i_*, and neglecting the linear and constant terms that do not arise in the problem at hand, a QCQP in general can be stated as (3)$$\begin{array}{c} minimize\;x^{H}Qx\\ subject\;to\;\left(s.t.\right)\;\;\;x^{H}S_{i}x\leq c_{i}, \end{array}$$

As long as ***Q*** and ***S**_i_* are psd, the problem is convex and can be readily solved. While the matrices involved in RF heating applications are indeed psd, we did not seek to minimize RF power deposition but to maximize it. Accordingly, ***Q*** in Equation (3) is replaced by the negative semidefinite matrix −***Q***, which renders the problem non-convex and thus hard to solve, with the general QCQP being in the class of non-deterministic polynomial-time (NP)-hard problems [42].

Please note that all spatial field components were explicitly added in the calculation of the power correlation matrices since the aim was to simply maximize power deposition, irrespective of the responsible field components. If for any reason direct control over power deposition caused by individual field components is desired, the integration in Equation (2) can be performed individually for each component, yielding three distinct matrices to allow directly targeting or constraining individual vector field components. Similarly, the extension to field magnitudes squared in lossless regions for general field shaping applications is straightforward [43].

### 2.2. Semidefinite Relaxation

QCQPs appear in a wide range of fields, and one of the most popular approaches to solve them is by applying a semidefinite relaxation [42]. In a first step, a trace operation (tr) is applied to the involved quadratic forms. Since the trace of a scalar is an identity operation, and using the cyclic property of the trace, any quadratic form expression can be reformulated to (4)$$x^{H}Qx\;=\;\mathrm{tr}\left(x^{H}Qx\right)\;=\;\mathrm{tr}\left(xx^{H}Q\right)\;=\;\mathrm{tr}\left(XQ\right)$$
where the rank 1 psd matrix ***X*** is formed as the outer product of the excitation vector *x* with its Hermitian transpose *x^H^*. The optimization problem now reads (5)$$\begin{array}{c} \min\mathrm{tr}\left(XQ\right)\\ s.t.\;\mathrm{tr}\left(XS_{i}\right)\leq c_{i}\\ \;X\geq 0\;\left(X\;\mathrm{is}\;\mathrm{psd}\right)\\ \;\mathrm{rank}(X)=1. \end{array}$$

The non-convexity originates from the rank constraint, and dropping this transforms the problem into a convex semidefinite program that only has one (global) minimum [44] that can be easily found using appropriate software tools. Among the most popular are the freely available SeDuMi [45], SDPT^3^ [46,47], SDPA [48,49] and CSDP [50]. Commercial solvers such as MOSEK [51] can offer significantly reduced computation times for large problems. Using these different solvers is conveniently facilitated by high-level modeling interfaces such as YALMIP [52] or CVX [53].

The remaining difficulty is the retrieval of a solution to the original problem from the relaxation. In case the solution rank of ***X*** is 1, the solution to the relaxed problem is also the solution to the original problem. *x* can then be retrieved from ***X*** using an eigendecomposition that will only yield one non-zero eigenvalue *λ* and its associated eigenvector *v*. The desired solution vector is then given by $x=\sqrt{\lambda }v$.

In general, the solution is of rank > 1, in which case additional steps are required to get to a feasible rank 1 solution [42]. However, in the case of localized RF heating, it can be shown that the full rank solution is of great value and has an actual physical meaning.

### 2.3. Time-Multiplexed Radiofrequency (RF) Heating

We now consider the case where not one but *m* distinct excitation pulses are played out in succession. The power P in this case evaluates to the average RF power delivered in each excitation: (6)$$P\;=\;\frac{1}{m}\overset{m}{\underset{k\;=\;1}{\sum }}x_{k}^{H}Qx_{k}$$

We can again take the trace of the expression and use the trace’s linear mapping property to arrive at (7)$$\begin{array}{c} tr\left(\frac{1}{m}\overset{m}{\underset{k\;=\;1}{\sum }}x_{k}^{H}Qx_{k}\right)\;=\;\frac{1}{m}\overset{m}{\underset{k\;=\;1}{\sum }}tr\left(x_{k}^{H}Qx_{k}\right)\;=\;\frac{1}{m}\overset{m}{\underset{k\;=\;1}{\sum }}tr\left(QX_{k}\right)\;=\;\frac{1}{m}tr\left(Q\overset{m}{\underset{k\;=\;1}{\sum }}X_{k}\right)\\ \;=\;\frac{1}{m}tr\left(QY\right) \end{array}$$

This already looks very similar to the previously shown expression for the semidefinite relaxation approach (Equation (4)). The matrix ***Y*** is the sum of the outer products of all excitation vectors with their Hermitian transpose: (8)$$Y\;=\;\overset{m}{\underset{k\;=\;1}{\sum }}X_{k}\;=\;\overset{m}{\underset{k\;=\;1}{\sum }}x_{k}x_{k}^{H}$$

Several observations about ***Y*** can be made. First, ***Y*** is psd by construction. Second, depending on the number of excitation vectors *m* and whether the individual *x_k_* are linearly dependent, the inequality 1 ≤ rank(***Y***) ≤ *N* always holds true, as a Hermitian matrix of dimension *N* can be of rank *N* at most. This means, that for any number of time-multiplexed arbitrary excitation vectors, we can find at most *N* alternative excitation vectors *u_k_*, resulting in an identical power deposition. With *u_k_* being the orthonormal eigenvectors of ***Y*** and *λ_k_* the associated eigenvalues, ***Y*** can be written as a weighted sum of outer products from its eigenvectors:(9)$$Y\;=\;\overset{N}{\underset{k\;=\;1}{\sum }}\lambda_{k}u_{k}u_{k}^{H}$$

These results of the time-multiplexed excitation case allow us to reach the following conclusions:A rank >1 solution to the semidefinite relaxation corresponds to the time-multiplexed excitation scenario.The individual excitation vectors for the time-multiplexed application can be retrieved using the eigendecomposition of ***X***.Any arbitrary number of excitation vectors can be effectively compressed to at most *N* vectors.

The possibility of compressing an arbitrary number of excitations into at most *N* vectors is not a new result, but is already leveraged for the rapid evaluation of local SAR for complex excitation pulse sequences involving hundreds of segments in parallel transmission magnetic resonance imaging (MRI) [54,55,56].

The semidefinite relaxation of the original problem (i.e., dropping the rank = 1 constraint) thus corresponds to a closely related problem with an actual physical meaning: the time-multiplexed RF heating scenario. We may not have gotten a free lunch [57], but an *N*-course dinner menu instead.

### 2.4. Arbitrary Heating Patterns

At this stage, we have arrived at a convex optimization problem to find the time-multiplexed excitations to maximize RF power deposition averaged over a chosen volume. For applications such as heating of large or multiple disjoint regions, it is conceivable that this might lead to solutions where some regions receive insufficient power deposition and others experience strong focal heating. It would be more desirable to directly minimize the deviation of local heating from a desired target pattern. Given *M* distinct heating target locations, we define a column vector *t*, whose entries are the local power depositions at the desired locations: (10)$$t_{i}\;=\;\mathrm{tr}\left(XQ_{i}\right),\;i\;=\;1\ldots M$$

The matrices ***Q**_i_* are the power correlation or SAR matrices of the target regions. Additionally, we defined a target power vector *r*, whose entries represent the desired local power deposition and a diagonal matrix ***W***, where the (*i*,*i*) element contains a weighting factor between 0 and 1, which ranks the importance of the *i*th target region. The optimization problem can now be stated as a constrained norm minimization:(11)$$\begin{array}{c} \min\;||W\left(t-r\right){||}_{p}\\ s.t.\;\mathrm{tr}\left(XS_{i}\right)\leq c_{i}\\ \;X\geq 0 \end{array}$$

As the norm is a convex function, this problem is again convex and can be tackled using semidefinite programming. Whether the weighted mean, sum-of-squares, or maximum deviation is to be optimized can be chosen by selecting the appropriate norm (*p* = 1,2 or ∞). The target points are not required to be densely distributed throughout the target volume, as the smallest focusing sphere has an approximate diameter of *λ*/3, with *λ* being the RF wavelength inside the medium [34]. Instead, the spatial distribution could be appropriately undersampled based on the local tissues and applied frequency.

### 2.5. Frequency-Multiplexed RF Heating

If an applicator capable of delivering RF power over multiple frequencies is available, the question arises of which frequency (or frequencies) would be optimal for a specific heating target, and if the concurrent application of multiple distinct frequency fields leads to an improvement. The current optimization problem stated in Equation (11) can be easily augmented to the multiple-frequency case. Since electromagnetic fields at different frequencies do not interfere, their respective heating patterns inside the target and healthy regions are entirely additive. Using *f* as a discrete frequency index variable, the multi-frequency heating vector *t_F_* is defined as(12)$$t_{F}\;=\;\overset{F}{\underset{f\;=\;1}{\sum }}t\left(f\right)$$
where *F* distinct frequencies are used and *t*(*f*) is the heating vector at frequency *f*. A similar summation is used to construct the multi-frequency constraint expression(13)$$s_{i}\;=\;\overset{F}{\underset{f\;=\;1}{\sum }}\mathrm{tr}\left(X_{f}S_{i,f}\right)$$
with *F* distinct optimization matrix variables ***X**_f_*, and ***S**_i,f_* representing the *i*th constraint matrix at frequency *f*.

The complete optimization problem now reads (14)$$\begin{array}{c} \min\;||W\left(t_{F}-r\right){||}_{p}\\ s.t.\;s_{i}\;\leq c_{i}\\ \;X_{l}\geq 0\; \end{array}$$
and fully describes the time-multiplexed multi-frequency constrained targeted RF heating problem. The solution will intrinsically determine which frequency (or frequencies) are most advantageous, and whether time-multiplexing multiple excitations (determined individually for each frequency) lead to the best approximation of the desired target pattern.

### 2.6. Iterative Solution

For a practical application of the derived formulation, the following steps are required:EMF simulation of the RF applicator with an appropriate model of the object under investigation for the desired frequencies.Calculation of appropriately averaged SAR matrices for regions targeted for RF heating and for regions outside the target region for all frequencies.Solution of the optimization problem using the calculated SAR and target matrices.Retrieval of the individual excitation vectors for each frequency.

While the first two steps can be readily approached using modern computational electromagnetic field simulation packages, finding a solution can prove computationally demanding. The main obstacle to overcome is the tremendous number of constraint matrices. SAR matrices are usually calculated on a three-dimensional voxel grid with a resolution between 1 and 5 mm, resulting in between 10^5^–10^7^ distinct matrices for each individual frequency. Solving an optimization problem of this magnitude is out of reach for readily available computing workstations and available semidefinite programming solvers. Different approaches have been suggested to tackle this such as dedicated algorithms incorporating highly parallel SAR calculation algorithms [56,58,59], or compression algorithms to reduce the number of constraint matrices to a smaller set of virtual observation points (VOPs) [60,61,62]. Notwithstanding this progress, the former is incompatible with readily available semidefinite programming solvers, while the latter suffers from a significant one-time computational burden to calculate the compressed matrix set, which will be further exacerbated once multiple frequencies have to be considered. Additionally, the intrinsic overestimation of the VOPs will produce non-optimal results.

Here, we propose an alternative approach that can readily deal with very large constraint sets, yielding the optimum solution in a small number of fast iterative calculations, with a runtime only weakly dependent on the total number of SAR matrices in the model. It is intuitively clear that not all constraints will be active at the optimum solution, as not all off-target voxels will experience the same local heating. It is instead more probable that a small number of “hot-spots” will act as the active constraints. The idea at the core of our approach is to iteratively assemble this set of active constraints required for an optimum solution, while leaving out most unused constraints.

The iterative solution algorithm is outlined in Figure 1. The choice of the initial subset and how many of the violated constraints to add to the next iteration is somewhat heuristic and has an impact on the total runtime. A good initial choice are all voxels that exhibit the strongest SAR values when heating with a specific single element, averaged over all frequencies. This results in *N* initial constraint matrices, which effectively act as a sort of individual channel forward power constraint. Additionally, we have found that adding the top *N* strongest violated constraints to be a good tradeoff between the number of iterations required and solver speed for a single iteration. In the constraint removal step, we considered a constraint for removal if it was unused in the previous three iterations.

### 2.7. Retrieval of Excitation Vectors

Once the optimization has concluded, the excitation vectors required to achieve the calculated RF hyperthermia result need to be extracted from the optimizer output. Recalling the formulation introduced in Equation (12) ff., an optimization using *N* sources each operating at *F* distinct frequencies will yield *F* distinct (*N* × *N*) psd matrices ***X**_f_*.Perform an eigendecomposition of each matrix ***X**_f_*, each yielding *N* eigenvectors ***v**_k_* and their associated eigenvalues *λ_k_*. Very often, the solutions will be strongly rank-deficient, having only a few large eigenvalues. Each individual excitation vector is given by $v_{k}\sqrt{\lambda_{k}}$.Compute the local SAR distribution for each of the *N* · *F* excitations and evaluate their respective influence on the target region (e.g., by calculating their maximum and mean SAR inside the target region).Discard all excitations that do not significantly contribute to the solution (e.g., all excitations whose maximum SAR contribution to the target region falls below a certain threshold). In our examples, we chose to discard all vectors contributing less than 0.1% to the overall solution.Scale the remaining vectors for time-multiplexing. If *M* solutions belonging to the same frequency remain, this indicates that time-multiplexing is required (i.e., the excitations are played out in succession during the application and each solution vector needs to be scaled by $\sqrt{M}$. Excitations at different frequencies do not interact coherently and can in principle be played out concurrently (i.e., their SAR patterns are purely additive). If the different frequency solutions are also applied in a time-multiplexed fashion, a similar scaling needs to be performed.

Once the solution vectors are extracted, the time-averaged power required for each excitation can be calculated via their norm. It should be noted that the algorithm does not prescribe how the time-averaged power is to be reached. There is a degree of freedom in deciding, for example, whether two time-interleaved excitations are played out for the same duration with the computed level of total forward power, or if the identical forward power is prescribed for both excitations and different application durations for the respective pulses are used to reach the required relative power levels.

### 2.8. Implementation and Validation

The described algorithm was implemented in MATLAB (The Mathworks, Natick, CA, USA) using the high-level optimization modeling toolbox YALMIP [52]. This toolbox allows one to directly formulate the objective function and constraints as they are stated in the equations, and automatically translates them into a format that can be parsed by a chosen low-level semidefinite programming solver. In our work, we utilized MOSEK, which offers a significant speedup for large problem sizes compared to other solvers. All optimization tasks were performed on a Workstation computer with a 10-core Intel Xeon W-2155 processor and 256 GB of RAM. Typical memory footprints of the performed optimizations stayed significantly below the maximum available RAM, requiring about 32 GB at most.

The proposed algorithm was applied to two different setups in order to demonstrate its features and help intuitively elucidate the “inner workings” of the algorithm. All simulations were performed using the time-domain solver of CST Microwave Studio 2020 (Dassault Systèmes, Vélizy-Villacoublay, France). The resultant field data were exported to MATLAB for further processing.

The first setup, shown in Figure 2, was comprised of a 2-dimensional circular water phantom irradiated by 32 broadband localized sources created by a plane wave incident on a small aperture.

A 2-dimensional model was recreated in the 3D domain by using periodic boundary conditions on the z-directed boundaries. The spatial mesh resolution was set to 0.75 mm. Water was modeled using the single-pole dispersion model provided by CST. The plane wave excitation bandwidth spanned 500–2000 MHz, with steady state frequency domain monitors recording fields at 100 MHz steps within this range. The time-domain convergence criterion was set to −80 dB. Local SAR matrices were calculated on the computational grid of 0.75 mm and subsequently rebinned to a 2.5 mm grid using a locally and globally energy conservative algorithm [63], thus reducing computational burden without sacrificing accuracy. All shaped heating computations were carried out on this grid without spatially averaging the SAR over multiple adjacent voxels.

This synthetic example was chosen because the low-loss pure water sample has the advantage of allowing deep field penetration over a wide range of frequencies, which will serve as a benchmark to demonstrate the main features of the algorithm. Both single- and multi-frequency optimization for simple circular as well as more complex shapes will be demonstrated.

The second setup involves the realistic high-resolution human voxel model MIDA [64], which differentiates 153 different structures at a spatial resolution of 500 µm. For excitation, a helmet-like structure enclosing the upper head of the head model was designed. Two different tumor geometries were embedded in the voxel model as heating targets for analysis. The first geometry features a single spherical tumor (r = 2 cm) approximately centered in the applicator. For the second geometry, two disjoint spherical tumors of different radii (r_1_ = 3 cm, r_2_ = 2 cm) were embedded at off-center locations. The simulated setup is shown in Figure 3.

The grid-like structure contains 40 sources, which can be driven independently and generate varying current distributions on the conductors. The mesh-like structure of the helmet applicator was inspired by similar approaches that are utilized to help design optimum RF coil arrays for MRI [65]. Excitation was performed with broadband pulses (100–1000 MHz) and steady-state fields again recorded in 100 MHz steps within the range. The simulation mesh resolution was variable, with a minimum mesh step size of 2 mm. Tissue dielectric properties were simulated using dispersive Cole–Cole models taken from the IT’IS database [66]. The tumor properties were also modeled as dispersive based on the literature data of human glioma [67]. Since the available literature did not contain any data for glioma above 500 MHz, dielectric properties were extrapolated to 1000 MHz. For the extrapolation, a mixed model of grey matter, white matter, and cerebrospinal fluid was fitted to the tumor dielectric properties between 100 and 500 MHz, which was then used to extrapolate up to 1000 MHz. This approach is motivated by preserving the relative dielectric behavior of the tumor compared to its surrounding tissues, which was found to have a 30% higher conductivity and permittivity compared to surrounding tissues between 5 and 500 MHz [67].

Local SAR matrices were again computed directly on the computational mesh and subsequently rebinned to an isotropic 5 mm grid, resulting in approximately 30,000 matrices per frequency sample (300,000 in total). SAR matrices for the tumor were extracted from this unaveraged dataset. To generate the healthy constraint matrices, all SAR matrices belonging to the tumor were set to zero to create a dataset containing only power deposition in healthy tissues. Subsequently, these matrices were convolved with a 10 cm^3^ spherical averaging kernel [68] as an approximation to spatially averaged 10 g SAR [19]. Voxels containing more than 10% air after the averaging procedure were discarded.

Targeted RF hyperthermia optimization was performed on the single tumor model for both single- and multi-frequency applications. For the two-tumor model, the multi-frequency solution with different relative weightings between the two tumors was explored. The results were quantified with SAR statistics over the tumor volume along with the TC25, TC50, and TC80 values, which detail the fraction of the tumor enclosed in the 25%, 50%, and 80% isolines of peak SAR, respectively [69,70,71,72].

For comparison with an established algorithm, we computed solutions to the single-tumor case using an implementation of the focused constrained power optimization (FOCO) algorithm [37] and compared it to the MVFS approach. For this purpose, we selected the central tumor voxel as the target and performed a rank-1 approximation of the corresponding SAR matrix. Any unaveraged (point) SAR matrix has a maximum rank of 3 as it is composed as a sum of the SAR matrices calculated from each of the field components in the x, y, and z direction. Hence, performing a rank-1 approximation is equivalent to picking the dominating field direction as is done in FOCO. Using this matrix inside our optimization yielded the same result as FOCO, except for a possible global phase factor which is, however, irrelevant for the power deposition. The result was then compared to several increasingly complex MVFS scenarios using the full rank center matrix, a single matrix averaged over the whole tumor, and spatial shaping over the tumor volume with different target values and norms.

Extensive benchmarking was performed to demonstrate the performance of the algorithm using the single-tumor human model. The optimization was performed with a set of reduced channels (4–40 in steps of 4) and frequencies (1, 2, 4, 6, 8, and 10) in order to show the runtime dependence on these parameters. This parameter sweep was performed for full target sampling (257 points inside the tumor) and for 8-fold spatial undersampling (32 points). Finally, we compared the iterative approach to an optimization using all constraints at once. Computation times for all shown results outside the benchmark cases were also reported. For some selected cases, run times were compared to values reported in the literature for particle swarm optimization and FOCO [34] as well as a scalar field shaping case [43].

## 3. Results

### 3.1. Phantom Setup Using a 32-Channel 2D Applicator

Most of the algorithm’s properties can best be understood from the examples in the 2-dimensional setup. The excitation of a spherical region (r = 19 mm) inside the water phantom is detailed in Figure 4. The desired target SAR was arbitrarily set to 1000 W/kg in order to maximize power deposition. SAR outside the target was constrained to 40 W/kg. The first optimization in (a) was performed using only the 500 MHz source, aiming to minimize the sum-of-squares deviation (i.e., 2-norm) from the target. The strongest contribution came from a focused mode with a bright central spot, however the algorithm provided two additional modes that deposit power in the periphery of the target while having much lower deposition in the center. Being at the same frequency, these three modes would have to be played out in a time-multiplexed manner to achieve the overall power deposition pattern. The overall mean and peak SAR values reached were 147.1 and 246.9 W/kg, respectively. For the next example shown in (b), all frequencies between 500 MHz and 2000 MHz could be used. The algorithm identified four modes at four distinct frequencies (500–800 MHz) as the optimum solution. At 500, 700, and 800 MHz, the excitations provided strongly focal heating in the center, with the 600 MHz excitation adding only very little SAR in the periphery. Mean and maximum SAR increased to 178.2 and 379.9 W/kg, corresponding to an improvement over the single-frequency result of 21 and 54%, respectively. The explanation for the improved performance is marked with arrows in (a) and (b): At 500 MHz, a circular region outside the target is mostly spared from power deposition. The higher-frequency excitations can “squeeze” their power deposition outside the target into this region, generating a complementary SAR distribution on the outside while simultaneously adding up their contributions inside the target. This is a key behavior of the proposed approach and one of its main working principles. Whereas the 2-norm was minimized in the first two examples, the optimization for (c) targeted the worst-case deviation (i.e., infinity-norm) from the target. While the mean value was about on par with the single-frequency result (a), the standard deviation of the local SAR inside the target was markedly reduced from 49.3 to 19.8 W/kg. Power was deposited much more homogeneously throughout the target, albeit at the cost of a reduced peak value. This homogenization is enabled by utilizing six excitations over four different frequencies. In this case, the excited modes showed complementary behavior both inside and outside the target. At both 1200 and 800 MHz, two modes with an almost orthogonal pattern emerged, offering high power deposition at complementary positions inside the target “avoiding” each other’s power deposition on the outside. It is important to note that not all the resulting excitations need to be used in the final application. The progression in (d) shows the cumulative effects of the first five modes identified as part of the solution in (c). While power deposition increases and homogenizes with each added mode, diminishing returns are to be expected at a certain point.

In principle, arbitrarily complex shapes with varying target values can be approximated, as is demonstrated in Figure 5. Here, the target SAR was chosen to be composed of 60 and 80 W/kg regions, which was lower than the previous example and allowed for trading off peak and mean power deposition for homogeneity [16]. While the target distribution (a) featured sharp edges, the weight distribution (c) was smoothed using a lowpass filter in order to reduce the impact of sharp edges, similar to that typically done in shaped excitation in the context of MRI [73]. This allows the algorithm to more strongly deviate from the target in the boundary regions of the desired shape. Together with an exclusion zone around the target as shown in Figure 4, this allows boundary regions of the target to be treated with flexibility. The achieved shaped power deposition pattern shown in (d) was almost perfectly flat, while most of the outside regions were very close to or at the constraint limit of 40 W/kg. As in previous examples, this is due to the complementary effects of multiple time- and frequency-multiplexed modes. A total of 43 different excitations was required to generate the shown pattern, with the first nine shown in (e). While this example is rooted in a Gedankenexperiment, it conveniently demonstrates the versatility of the proposed approach.

### 3.2. Human Brain Model Setup Using a 40-Channel 3D Helmet Grid Applicator

Moving from the realm of instructive academic examples to scenarios more closely resembling the intended application of the proposed algorithm, the results from the targeted RF heating simulations using the human head model with a single central tumor are shown in Figure 6. Using only one frequency (600 MHz), the resultant target pattern is shown in (a) and (b). Strong focal heating becomes evident with a mean SAR in the tumor of 76 W/kg, an increase of a factor of 1.9 over the allowed healthy tissue SAR of 40 W/kg. As before, the individual contributing modes are shown in a different color map in (c,d). Again, both modes contribute in the center of the target region and occupy complementary regions inside the healthy tissues. To elucidate the electric vector fields belonging to these power deposition patterns, the computed excitation vectors were imported into CST and used for result combination, which allows for visualization of the associated fields, as is shown in (e,f). The two power deposition patterns are generated by two counterrotating circularly polarized electric fields inside the tumor region, with the rotation direction indicated by the two circular arrows. This is a notable distinction to previous approaches [35,37], which rely on picking a dominating field component and performing the optimization using only this direction. Not only are multiple electric field components mixed in one resulting mode of the proposed approach, but multiple polarizations are identified as complementary excitations to achieve an improved target heating pattern.

Allowing all frequencies between 100 and 1000 MHz to contribute yielded the heating pattern shown in (g–l). The overall performance was only slightly increased compared to the single frequency optimization.

The results for the two-tumor model are summarized in Figure 7. At first, RF induced heating was individually optimized for each tumor, which is shown in (a) and (b). For each target, different optimal frequencies were determined. This is due to the size and location of each tumor, with the larger tumor having the dominant mode at 400 MHz as opposed to 500 MHz for the smaller tumor. The second strongest contribution to the large tumor model lay at 700 MHz and 600 MHz for the small tumor. This indicates that size and tumor location are important for the optimum RF heating frequency, with higher frequencies supporting smaller focal spots at the cost of a reduced penetration depth.

For concurrent heating of both tumors, the results are shown in rows (c)–(e). For equal weighting (c), the large tumor was strongly favored, which intuitively makes sense because it contributes more voxels to the optimization and is thus more highly weighted on average. Increasing the relative weighting of the smaller tumor by a factor of 2 equalized the relative peak SAR in both tumors (d) whereas a relative weighting of 3 shifted the focus more strongly to the smaller tumor (e).

In general, it can be observed that the applied power is not “smeared out” over the whole available frequency spectrum, but rather that one or very few distinct frequencies have a strongly dominant contribution.

### 3.3. Runtimes of Demonstration Examples

The runtimes and corresponding solver parameters for all shown examples are summarized in Table 1. The benchmarked 2D setups required a higher number of iterations and constraints compared to the 3D human model examples. This is because RF power dissipation inside the water phantom is much more uniform due to the lack of any dielectric interfaces. This results in a larger set of active constraints at the optimum solution, which accordingly takes longer to assemble. The shaped logo excitation required the longest computation time due to the high number of target points and frequencies employed in the example. For all human model examples, significantly less than 1% of all possible constraint voxels were active at the solution, indicating that only a few hotspots limit the power deposition.

### 3.4. Iterative vs. Non-Iterative Approach

The single-tumor problem with 10 frequencies and 40 channels was solved in 10 iterations using 230 out of the total 30,000 constraint voxels. Not relying on the iterative algorithm, but directly using all 30,000 constraints prolonged the solution time from 204 s to 5290 s, an increase by a factor of 26. Using larger or more highly resolved models, the non-iterative solution time would become even more prohibitive. The iterative solver on the other hand can deal with larger models. Increasing the rebinning resolution from 5 mm to 2.5 mm with a corresponding 8-fold increased the number of constraint voxels (240,000), the iterative solver finished after 17 iterations in 430 s. This constituted a 2-fold increase in runtime, indicating a significantly sublinear runtime scaling with model size. To keep all other parameters constant and allow a fair comparison, the target volume was undersampled by a factor of eight to maintain the same number of target points.

### 3.5. Dependence on Frequency and Channel Number

The result of the parameter sweep used for the evaluation of the impact of channel and frequency number on the optimization runtimes is summarized in Figure 8. The required time increases approximately with *N*^3^ − *N*^4^ and shows an almost linear dependence on the number of frequencies F. Due to the heuristic nature of the iterative solver, the performance graph is not entirely monotonic. Reducing the number of target points by a factor of 8 decreased the runtimes by 30% on average.

Based on these results, a rough performance comparison to other approaches can be attempted. Bellizzi et al. reported average optimization times of 21.6 s for FOCO and 36.6 s for a particle swarm global optimization approach [34] using a 20-channel array for the treatment of head and neck tumors, of which 12 active elements were selected during hyperthermia planning. A single frequency 20-channel optimization required 2.5 s using MVFS, whereas a 12-channel run finished in merely 0.7 s. A scalar field shaping example using two target points and 96 single-frequency sources was reported to take 2 h to compute [43]. Extrapolating from the data shown in Figure 8 for a single frequency, MVFS would require approximately 12–13 min for a problem of similar complexity. It should be noted that the performance comparison is in no way definite due to the large number of unknowns having to be considered for a proper performance evaluation such as implementation details, available hardware, exact problem size, etc.

### 3.6. Comparison of Multiplexed Vector Field Shaping (MVFS) to Focused Constrained Power Optimization (FOCO)

The comparison between our FOCO implementation and multiple MVFS scenarios are summarized in Table 2. All computations were done using all 40 channels at 600 MHz since this frequency was identified as the most efficient for the target volume. Compared to using the dominating field component in FOCO, just utilizing the full-rank central SAR matrix (“Center” column) already provides an increase in power deposition with mean, maximum, and minimum SAR increased by 9%, 12%, and 24%, respectively. Tumor coverage TC50 was slightly decreased, however, this coverage metric was measured relative to peak SAR and the increased maximum value led to a decrease in TC50. The direct field shaping examples, denoted with “S”, demonstrate the capability of MVFS to distribute power throughout the target more uniformly. As an example, using the infinity norm with a target SAR of 150 W/kg (“S_∞_150”) achieves the same mean SAR as FOCO, but raised the minimum value inside the target by 55% and increased TC50 from 0.87 to 0.95. All MVFS results differed over the given metrics, which would allow for weighing different treatment aspects and choosing the optimum result for the given treatment scenario. As demonstrated in Figure 8, computation times for current clinical channel numbers were shown to be 2.5 s for 20 channels and 0.7 s for 12 channels. These short runtimes conveniently allow solving for multiple target patterns to iteratively approach a power deposition pattern that is expected to perform best.

## 4. Discussion

This work introduces and evaluates the multiplexed vector field shaping (MVFS) approach, a convex formulation of the time- and frequency-multiplexed constrained RF heating problem without unwarranted simplifications and provides an iterative algorithm to efficiently find its globally optimum solution. It was shown that allowing multiple time-interleaved excitations removes the non-convexity arising in the classical RF heating problem, which seeks only one single superposition. For an *N*-channel array, up to *N* distinct excitations per operating frequency can theoretically contribute to the heating pattern, however, the solutions found for realistic human body models have been typically of low rank with only two or three contributing modes. Our results demonstrated the validity of the derived method using a 2D circular water phantom irradiated by 32 broadband localized RF sources. While not representing a direct clinical application, the excitation of a highly detailed shape within the phantom serves as a proof-of-principle demonstration of the algorithm.

Given appropriate RF applicators, we demonstrated that even complex regions can be excited using a 3D human brain model including a spherical tumor in conjunction with a 40-channel broadband helmet RF applicator. Using the relative weighting feature, it was shown that an optimum balance for targeting disjoint or intricately shaped regions can be determined to equalize RF power distribution. Further studies are required to establish a guideline on how to best choose a relative weighting for disjoint or otherwise complex regions.

Extensive performance benchmarks have demonstrated the applicability of MVFS for time-critical applications. Typical clinical problem sizes using a single frequency and a maximum of 20 channels were solved in 2.5 s or less, lending the algorithm to an explorative approach for finding the best power deposition pattern for the envisioned treatment plan. At the other end of the spectrum, extraordinarily complex scenarios using up to 10 different frequencies and 40 channels were solved in around three minutes or less, which allows for extensive design optimizations in an offline setting. The iterative solution procedure was demonstrated to significantly reduce the required computational workload. The presented approach is not limited to hyperthermia and supports RF induced thermal intervention planning for temperature ranges with T > 48 °C.

The proven global optimality of our approach primes it to be a valuable tool not only for RF hyperthermia planning, but also for the development and comparison of multiple applicator designs to improve upon the available hardware options. Previous approaches such as field focusing or FOCO [28,34,35] can be viewed as special cases of the presented general solution of the time- and frequency-multiplexed problem. For example, picking a single point with a dominating field component as the optimization target corresponds to only using a rank-1 approximation of the respective target SAR matrix in our approach. It was demonstrated that MVFS can provide multiple complementary excitations that effectively homogenize power distribution throughout the target compared to using only a single target point and field polarization.

Arbitrary target patterns and weights supported by our generalized approach provide numerous degrees of freedom to best deliver RF energy at the desired locations. In particular, vulnerable locations such as the eyes with high electrical conductivity can easily be protected by applying more strict constraints to local power deposition in these regions. If appropriate constraints are employed, the presented solution can also be used for modulation of the electric-field distribution of a RF array to generate reduced E-field zones or even electric field-free zones in the body without significantly altering the transmit sensitivity used for magnetic resonance imaging (MRI) [74,75]. This approach would facilitate the reduction of RF induced heating in passive, conductive implants through modification of the electric field of the transmit RF array. This would benefit the development of implant-friendly RF coils and promote MRI patient safety in the presence of (biodegradable) implants. It would also afford alleviation of absorbed RF power around deep brain stimulation devices in MRI [76]. Finally, the presented approach can be directly applied to temperature rise matrices instead of SAR matrices, which would permit a more direct approximation of the desired temperature distribution [23,40].

It is a recognized limitation of our approach that individual RF channel constraints for a single excitation such as limits on available power for a single RF channel cannot be enforced in a straightforward manner. A potential extension could be to constrain the maximum eigenvalue of the solution matrices ***X*** (i.e., its 2-norm), which comprises a convex constraint and is thus valid within the proposed framework. While this does not limit individual RF channel power, it nevertheless constrains total forward power over all channels during a single excitation. Additionally, since local SAR itself acts as a dampening factor preventing a single channel to be overly dominant, we do not expect overly unrealistic solutions to arise in other setups [77,78].

## 5. Conclusions

In this work, we have proposed, derived, and demonstrated an approach to quickly solve the time- and frequency-multiplexed RF heating problem to global optimality. This methodology, termed multiplexed vector field shaping (MVFS), provides a technological foundation for the advancement of RF hyperthermia as an adjunct anti-cancer treatment option including sophisticated treatment planning and safety management as well as the hardware design side. Since the field shaping question is also relevant for other disciplines [43], we anticipate that our approach also spurs innovation outside of our first targeted applications.

## Figures and Tables

**Figure 1 cancers-12-01072-f001:**
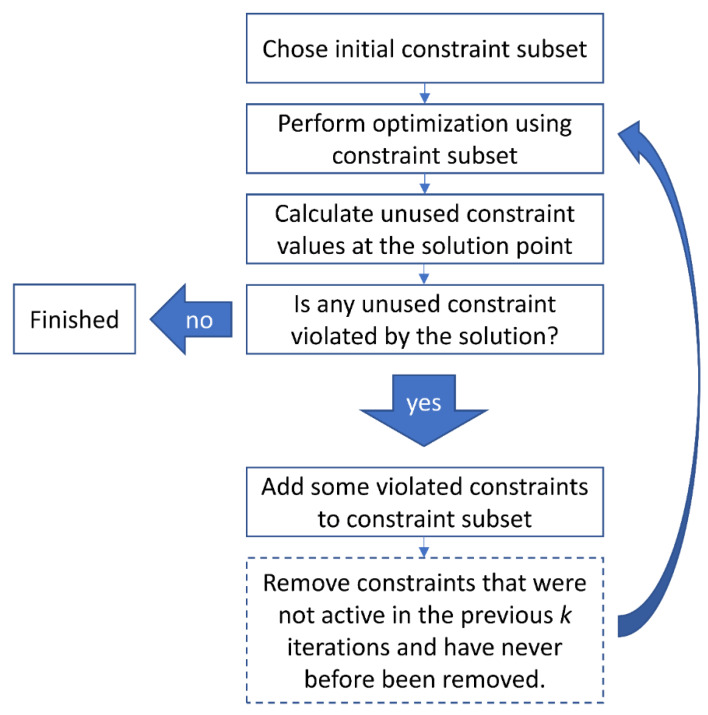
Flowchart of the iterative solution algorithm. The algorithm initially selects a small subset of all healthy (=constraint) voxels to be considered during the targeted heating calculation. After performing the optimization, the resultant specific absorption rate (SAR) in the unconsidered voxels is calculated to find regions where the found solution violates the constraints (i.e., leads to undesired heating in healthy regions). A small number of the healthy voxels experiencing the strongest heating are added to the constraint subset and the optimization is repeated. This process is iterated until no further constraints are violated by the solution. The last step in the dashed outline is optional and only required if the number of constraints has increased to a level that significantly impacts each iterative solution runtime.

**Figure 2 cancers-12-01072-f002:**
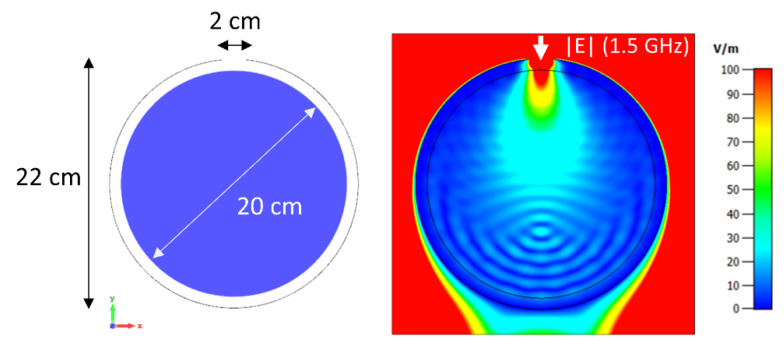
Depiction of the simulated 2D setup using a pure water phantom. A plane wave with E_Z_-polarization incident on a 2 cm aperture in a perfectly conducting shield is used as an excitation source to mimic a localized broadband radiofrequency applicator. In consecutive simulations, the aperture was rotated around the sample in 32 steps to provide completely circumscribing field sources. An exemplary E-Field plot is shown on the right.

**Figure 3 cancers-12-01072-f003:**
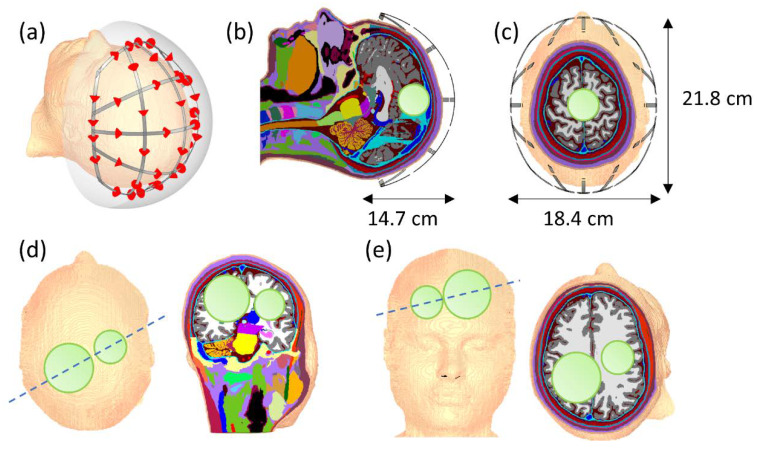
(**a**) Synthetic 40-channel helmet grid applicator for 3D evaluation of the proposed algorithm. The conductive paths conform to the contours of the head and can generate electromagnetic fields with varying polarizations. Simulated power sources are marked as red arrows. The RF applicator is shielded by a continuous conformal perfectly conducting shield positioned 2 cm away from the conductors (grey shaded area). (**b**) Sagittal and (**c**) coronal view of a spherical tumor (radius 2 cm) that was incorporated into the brain of the human voxel model and positioned at the center of the applicator (light green shaded area). (**d**) and (**e**) show oblique slices through the center of the second tumor model using two differently sized tumors (radius 2 and 3 cm).

**Figure 4 cancers-12-01072-f004:**
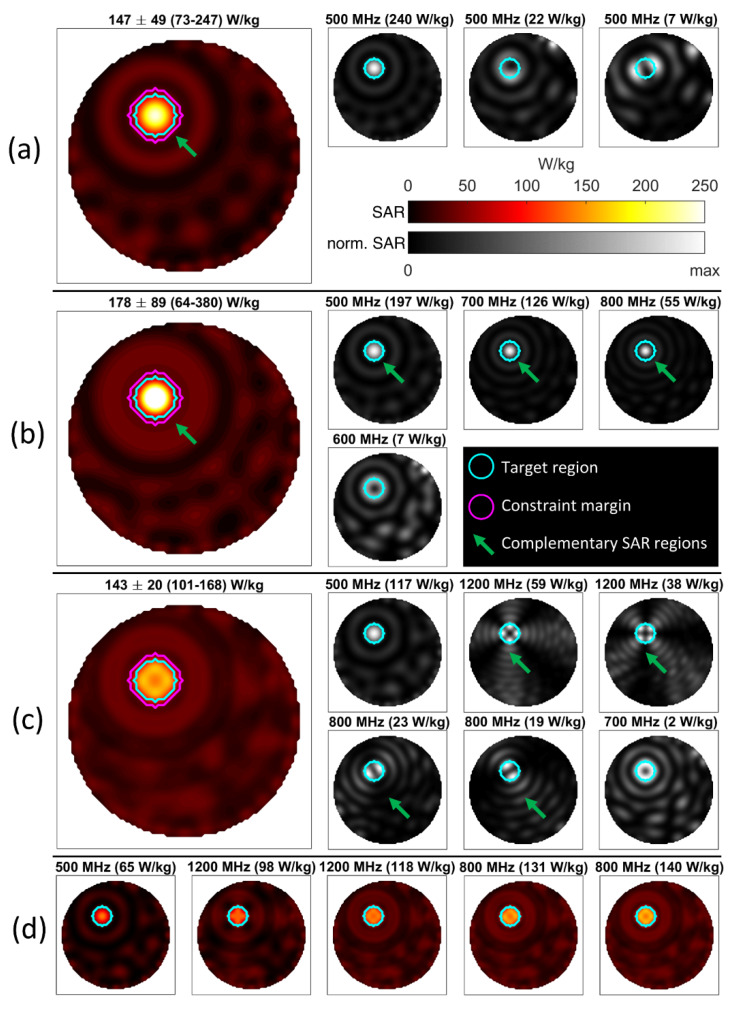
Demonstration of shaped RF power deposition in the 2D water sample. The target region is delineated with a cyan contour (r = 19 mm), and the region between the cyan and magenta contour indicates a “safety margin” where no constraints are enforced. (**a**) Optimization result using only 500 MHz sources minimizing the 2-norm. The large plot on the left details the total achieved local SAR deposition, with statistical measures of SAR within the target region detailed in the header (Mean ± SD (Min–Max)). The smaller plots to the right show the contributions of multiple time-multiplexed modes scaled to their respective maximum, with their respective peak contribution inside the target region shown in the header. In (**b**), all frequencies between 500 and 1500 MHz could be used, resulting in an improved power deposition pattern. The green arrows indicate a region where SAR in the constrained regions is spread out between the different frequencies, which demonstrates a key principle of the proposed algorithm. For (**c**), the optimization was set to minimize the worst-case deviation from the target. This resulted in a much more homogeneous power deposition, albeit with lower mean and peak values. Again, all excitations contributed to the target region but occupied complementary regions outside it as indicated by arrows. The image succession in (**d**) shows the cumulative effects of the first five excitations from (**c**) to demonstrate how the different patterns build up the final superposition. Here, the resulting mean value is given in the header.

**Figure 5 cancers-12-01072-f005:**
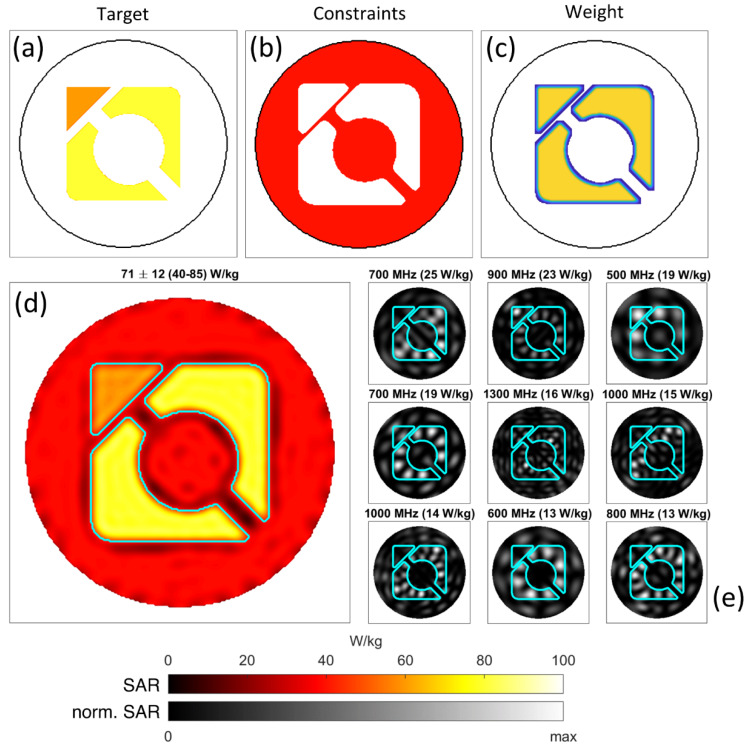
Excitation of a complex disjoint shape inside the water phantom (the logo of the first author’s affiliation). The top row shows the target pattern with 60 and 80 W/kg regions (**a**), constraints of 40 W/kg (**b**), and weighting distribution (**c**) used for the optimization. The achieved heating pattern is shown in (**d**), which requires time- and frequency-multiplexed excitation using 43 different modes between 500 and 2000 MHz. The first nine modes with the strongest peak impact are shown in (**e**). Measures of SAR within the target region are detailed in the header (Mean ± SD (Min–Max)).

**Figure 6 cancers-12-01072-f006:**
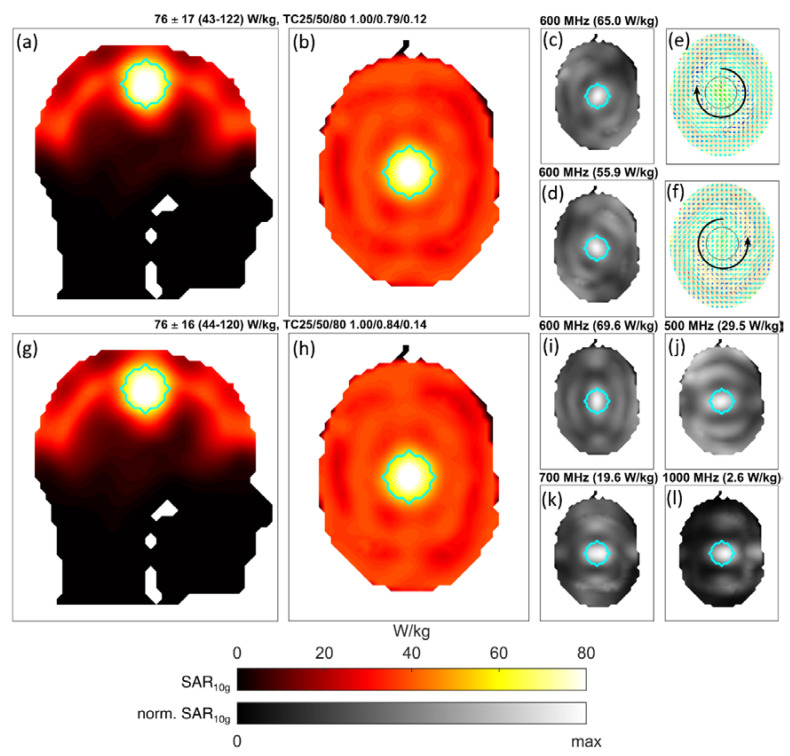
Results of the 3D heating example using the human head model. The target is again outlined in cyan and target SAR was set to 150 W/kg, minimizing the 2-norm of the deviation from this value. Measures of SAR within the target region are detailed in the header (Mean ± SD (Min–Max)), along with TC25, TC50, and TC80 values for the tumor. Images (**a**) and (**g**) show sagittal slices through the tumor center, whereas the other SAR plots represent axial maximum intensity projections over the whole volume. Images (**a**) and (**b**) show the achieved power deposition pattern when using only 600 MHz fields. This pattern is achieved by two time-multiplexed modes, whose patterns are shown scaled to their individual maximum in (**c**) and (**d**) analogous to the previous figures. These two modes correspond to two counterrotating circularly polarized electric fields inside the tumor. E-field vector snapshots and the rotation direction are shown in (**e**) and (**f**). This example demonstrates that the algorithm can arbitrarily mix polarizations within a single excitation (a circular polarization being comprised of two linear components) as well as yield differently polarized time-multiplexed complementary solutions. The second row of results in (**g**–**l**) shows the optimum result when allowing the use of all frequencies between 100 and 1000 MHz. From a target coverage standpoint, this solution performed only slightly better, with an identical mean but modestly lowered maximum, elevated minimum, and lower standard deviation.

**Figure 7 cancers-12-01072-f007:**
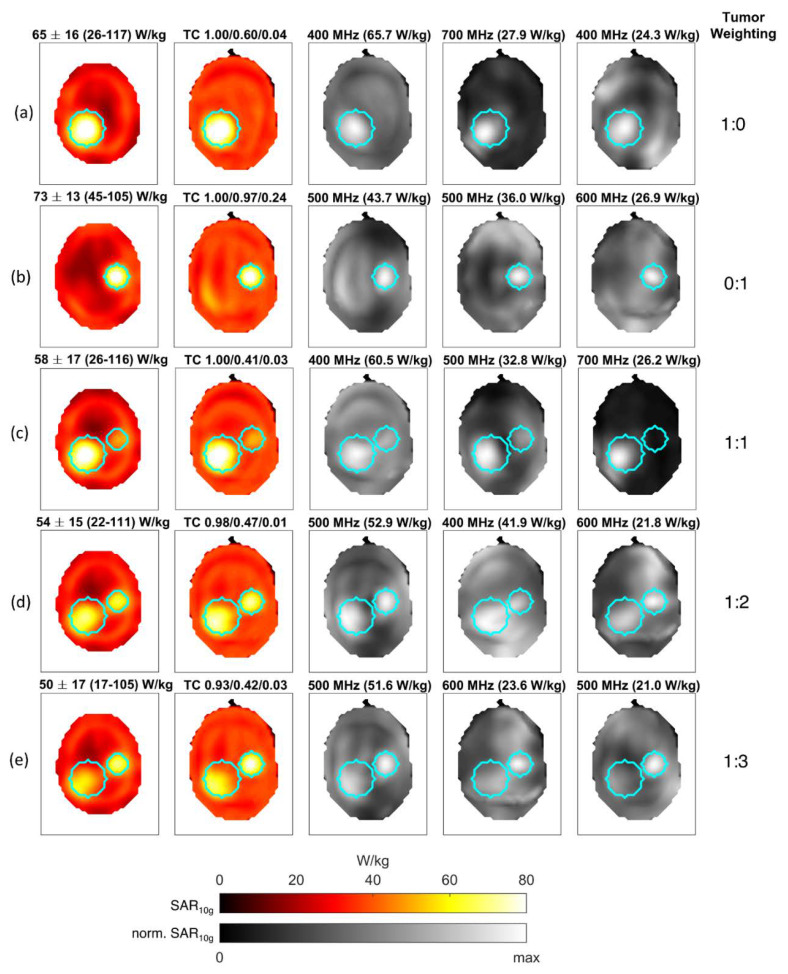
Results for the differently weighted two-tumor optimization. The leftmost image column displays axial slices through the target center; all other images are axial maximum-intensity projections with the results for individual contributing frequencies and modes being scaled to their individual maximum that is stated above them along with the frequency. Measures of SAR within the target region are detailed in the header (Mean ± SD (Min–Max)) along with TC25, TC50, and TC80 values for the tumor. Targeting each tumor separately leads to SAR patterns shown in (**a**) and (**b**). From (**c**) to (**e**), the weight of the smaller tumor increased from an equal to threefold weighting. Due to their different sizes, the target voxels belonging to the smaller tumor require a higher relative weighting for an approximately equal SAR deposition.

**Figure 8 cancers-12-01072-f008:**
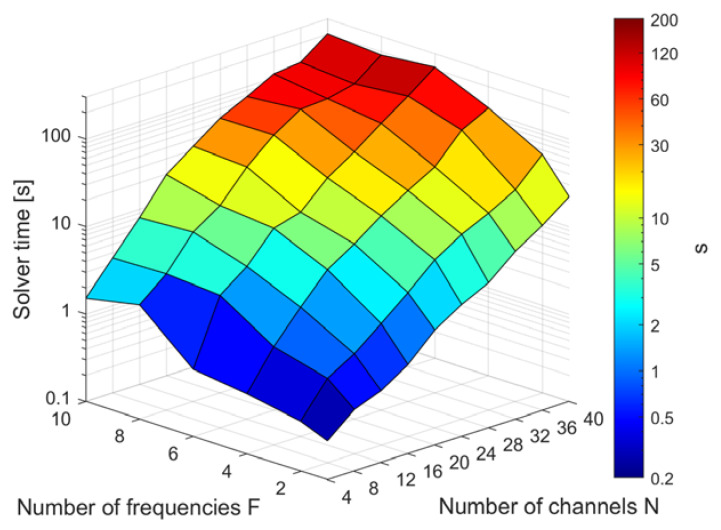
Runtime dependence on the number of available RF channels and frequencies used during the optimization. Since performance varied from the sub-second range up to about three minutes for the most complex scenario, a logarithmic scale was used. All calculations were performed by the iterative algorithm using the single-tumor model with 257 target points and approximately 30,000 total constraint voxels. For typical clinical problem sizes (single frequency, maximum of 20 channels during the planning stage [34]), the optimization times stayed below 2.5 s. The largest problem with all 40 channels and 10 frequencies was completed in 3 min and 24 s.

**Table 1 cancers-12-01072-t001:** Overview of the problem size and required computational time for all examples.

Example Figure #	# of Channels	# of Target Points	# of Frequencies	# of Constraint Voxels	% of Used Constraint Voxels	# of Iterations	Computation Time [hh:mm:ss]
4 (a)	32	149	1	4952	17.4	25	00:00:52
4 (b)	32	149	10	4952	15.3	10	00:03:21
4 (c)	32	149	10	4952	13.9	16	00:04:42
5	32	8693	16	22,289	1.7	19	02:36:18
6 (a–f)	40	257	1	29,914	0.7	9	00:00:22
6 (g–l)	40	257	10	29,914	0.8	10	00:03:24
7 (a)	40	247	10	29,908	0.6	8	00:03:10
7 (b)	40	67	10	29,908	0.4	9	00:02:52
7 (c)	40	314	10	29,908	0.6	8	00:03:04
7 (d)	40	314	10	29,908	0.5	9	00:03:16
7 (e)	40	314	10	29,908	0.6	8	00:03:03

**Table 2 cancers-12-01072-t002:** Comparison of multiplexed vector field shaping (MVFS) to focused constrained power optimization (FOCO) for the single tumor model using 40 channels and single frequency fields at 600 MHz. Results are separated into SAR statistics, tumor coverage, and solution details. FOCO was performed using a rank-1 approximation of the central tumor SAR matrix and compared to six different MVFS application scenarios. “Center” uses the full-rank central SAR matrix, “Averaged” utilizes a single target matrix built from averaging over the whole tumor volume. The remaining four examples used all 257 tumor SAR matrices to derive a field shaping (“S”) result. Here, the subscript defines the target norm used (either 2 or ∞) and the following number stands for the target SAR in W/kg (either 75 or 150 W/kg). The “Rank” row describes how many time-interleaved solutions were identified for the respective solution. As expected, the FOCO solution was of rank 1 while MVFS provided multiple excitations to better cover the target volume. The results for S_2_150 are visualized in Figure 6a–f.

Performance		MVFS						
Metrics		FOCO	Center	Averaged	S_2_ 150	S_2_ 75	S_∞_ 150	S_∞_ 75
Local 10 g-SAR [W/kg]	Mean	69	75	76	76	68	69	68
	Max	107	120	123	122	95	103	99
	Min	33	41	43	43	45	51	51
	SD	15	17	17	17	11	12	12
Coverage	TC25	1.00	1.00	1.00	1.00	1.00	1.00	1.00
	TC50	0.87	0.80	0.77	0.79	0.99	0.95	1.00
	TC80	0.13	0.14	0.12	0.12	0.23	0.16	0.18
Solution	Time [s]	16.5	14	22.2	22.8	21.7	24	26
	Rank	1	2	2	2	3	3	3

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
