# Peer review of "Solving the Time- and Frequency-Multiplexed Problem of Constrained Radiofrequency Induced Hyperthermia"

_cancers, 2020, doi:10.3390/cancers12051072_

Round 1
Reviewer 1 Report
The quality of the manuscript is very high. It represents an important step forward in hyperthermia treatment planning and in hyperthermia hardware design.
Here some minor suggestions to slightly improve the manuscript.
- Page 2 line 54: I suggest you also add a reference to Jacobsen, S., Melandsø, F. The Concept of Using Multifrequency Energy Transmission to Reduce Hot Spots During Deep-Body Hyperthermia. Annals of Biomedical Engineering 30, 34–43 (2002).
- In Eq. (14) on page 6, line 201, should the subscript "f" be changed into "F" ?
- At line 225 of page 6, the [36] references to a patent (please specify it on page 16 line 633, adding also the patent number that should be US 10,088,537 B2) that seems to not deal with a "highly parallel SAR calculation algorithms", but rather with the use of MR thermometry as a tool for hyperthermia treatment planning.
- On page 8 line 293, EZ must be probably corrected into EZ
- On page 8 line 298, are the boundary conditions on z-directed boundaries "periodic" or rather PEC? Please consider that in this example the E field has only the Z directed component.
- On page 14 line 472, the "(e)" should be changed into "(d)".
- Minor spell check is suggested. For example the sentence on page 16 line 517 says "It is a recognized limitation of our approach is that individual RF channel constraints..." where I bolded a word that should be removed.
Author Response
Thank you for the comments. Please find our replies below:
- We have added the reference along with others to the literature review.
- This is correct, we have fixed this in the revised manuscript.
- A part of the patent concerns the efficient parallelized calculation of SAR using a SAR “oracle”. For clarification, we have incorporated another publication into the references. The reference now also outlines the patent number.
- We have fixed this.
- We did set the boundary conditions in CST to “periodic”, having an infinitely extended setup in mind. In this case no E-Field components exist along the xy-plane, so this is probably equivalent to PEC which also forces tangential E components to zero.
- We have reworked this figure and corrected this mistake.
- This was fixed.
Reviewer 2 Report
The manuscript is well written and some minor points given below should be addressed to improve the readability of the text.
- Have the authors used the default parameters in the SEDUMI solver? Please state clearly in the text.
- Concerning the developed iterative algorithm in section 2.6, the authors proposed a heuristic technique to speed up the optimization problem. Please state clearly in the text the number of iterations employed in the algorithm (figure 1) in all the examples (the authors have only stated that it is small). In addition, state in the text the computational time required by the proposed algorithm compared to the one with the use of all constraints.
- The authors have presented an interest 3D model with two disjoint tumors. Is there a general guideline to select a priori the factor concerning the relative weighting to equalize the relative peak SAR?
Author Response
Thank you for the comments. Please find our replies below:
The solver parameters were left unchanged from the defaults used internally by YALMIP. For the revised version we have utilized MOSEK as the solver due to its higher performance. This is now clarified in the manuscript.
The number of iterations required for all shown examples is now given in the performance benchmark section of the manuscript. Following the referee’s suggestion we have also added a performance comparison between the iterative solver and one example using all constraints.
So far there is no general guideline available due to the still relatively low number of investigated scenarios. Due to the algorithm’s high performance, investigating multiple different weightings does not constitute a major drawback. We have added a clarifying statement to the discussion section.
Reviewer 3 Report
In this paper, authors introduce their new method to find globally optimal solution to the electric field focusing problem. Introduction is well written and the very promising results are given. While they report very impressive theoretical results, the technical merit in their approach is not clear. The method haven’t been benchmarked to any other method which can be at least done for single frequency excitation setup. The computational times of the optimization haven’t been discussed at all. Number of channels of the synthetic applicator used in this study are unrealistically high. A more realistic model would give better indications for the future prospects.
Specific comments:
- Page 6 Line 232: Authors mention in the required steps, EMF simulations and calculation of SAR is required. The alternative proposed approach is claimed to alleviate these drawbacks. It is not clear to me if the method proposes faster EMF simulations as well. Please clarify this part.
- Page 7 Line 273: It is not clear how the duration of a pulse is selected when the excitations are played out in succession during the application. Are they equal in length or are they also optimized?
- Page 9 Line 348: Please clarify in methods single- and multi-frequency applications were also applied for two tumor model.
- Methods/Results: Please add the computational times and the work station specifications.
- Please add the total run time needed to achieve the results presented in Figure 5. Please comment on its applicability in clinic in discussion.
- How is the global optimality proved?
- Are the SAR calculated for different frequencies combined coherently?
- Page 16 Line 495: Throughout the paper, the focus is on the SAR but here you make a claim regarding temperature. Please explain what is meant here, and support your claim.
- Page 13 Line 430: Please define mean elevation factor before using it in results.
- Please report TC20, TC50 and TC80 values for the results given in Figure 6 and 7 to be in line with other hyperthermia studies and homogenization of the report of the study results.
- Reporting an actual SAR distribution for a slice for head model (such as slice shown in Figure 3 d or e) would be beneficial. Maximum intensity projections smooth out the results and the behaviors at the tissue interfaces are not easily visible.
- Please clarify in the figure captions that the reported SAR values above the projections are mean values in the target region.
- Claims in the conclusion or in the abstract are weakly (or not) backed by quantitative results. Claims such as high performance, ideally suited for ideally suited for interactive hyperthermia treatment planning (what is the runtime?) are empty when the quantifications are not done. RF hyperthermia is already in use in the clinic as adjuvant as an adjunct anti-cancer treatment option. It is hard to see the merit for clinical use without quantitatively backed results.
Author Response
Thank you for your comments. We have incorporated an extensive benchmarking and comparison section to the paper to address the concerns. Please find our point-by-point response below. Our answers are marked in red.
In this paper, authors introduce their new method to find globally optimal solution to the electric field focusing problem. Introduction is well written and the very promising results are given. While they report very impressive theoretical results, the technical merit in their approach is not clear. The method haven’t been benchmarked to any other method which can be at least done for single frequency excitation setup. The computational times of the optimization haven’t been discussed at all. Number of channels of the synthetic applicator used in this study are unrealistically high. A more realistic model would give better indications for the future prospects.
The lack of performance benchmarks and comparisons is a valid concern and is addressed in the revised manuscript. Following the referee’s recommendation, a comprehensive section detailing the runtimes, required iterations etc. for all shown examples is added.
Secondly, the single-tumor model is used as a benchmark scenario to detail the scaling capabilities of the algorithm with regards to the number of target points, number of RF channels, number of frequencies and number of constraint voxels.
In order to give a rough estimation on how the performance compares to alternative approaches such as the focusing via constrained power optimization (FOCO) algorithm, we implemented a “FOCO-like” optimization for the single-tumor, single-frequency model within our framework. Additionally, typical FOCO and particle swarm optimization runtimes from the literature are mentioned in the text.
We are aware that the number of RF channels is high compared to currently used hyperthermia systems. However, the goal was to have a complex benchmark system providing diverse polarizations while covering the whole brain in order to demonstrate the algorithm’s capabilities, not to propose an applicator design. The mesh-like structure of the helmet applicator was inspired by similar approaches that are utilized to help design optimum RF coils for MRI (see newly added reference from Paska et al.). The computational performance of the algorithm on systems with lower channel numbers is now provided in the benchmark section.
Page 6 Line 232: Authors mention in the required steps, EMF simulations and calculation of SAR is required. The alternative proposed approach is claimed to alleviate these drawbacks. It is not clear to me if the method proposes faster EMF simulations as well. Please clarify this part.
Our algorithm still requires the same simulations and SAR matrix calculation as other approaches based on numerical EMF simulations. The performance benefit is gained purely in the optimization stage. We have clarified this point in the revised version of the manuscript.
Page 7 Line 273: It is not clear how the duration of a pulse is selected when the excitations are played out in succession during the application. Are they equal in length or are they also optimized?
This is an excellent point that requires clarification. The algorithm yields different power levels for each excitation. In the real application, different relative power levels between pulses can be achieved by either having different amplitude scalings and playing each excitation out for the same time or by varying the duration of each excitation according to their relative forward power. This is now explained in section 2.7 on the result vector retrieval.
Page 9 Line 348: Please clarify in methods single- and multi-frequency applications were also applied for two tumor model.
The section was reformulated accordingly.
Methods/Results: Please add the computational times and the work station specifications.
A detailed benchmark section as well as workstation specifications was added.
Please add the total run time needed to achieve the results presented in Figure 5. Please comment on its applicability in clinic in discussion.
The total runtime is now given in the performance benchmark section. This example is not directly applicable to a clinical setting but is intended as an “all-out” benchmark example to demonstrate the capabilities of the algorithm and allow the reader to visualize how even complex target shapes are approached using complementary spatial power deposition profiles. We have mentioned this in the discussion section of the revised version of the manuscript.
How is the global optimality proved?
The problem is carefully formulated as a semidefinite optimization problem which only has a single (global) minimum due to its convexity. We emphasized this in the manuscript and added a reference regarding the convexity of SDPs in section 2.2.
Are the SAR calculated for different frequencies combined coherently?
The electric fields at the different frequencies are, in our examples, separated so far that they do not interfere coherently. Hence their SAR patterns are purely additive. We have added an additional clarifying statement to address this question in section 2.7.
Page 16 Line 495: Throughout the paper, the focus is on the SAR but here you make a claim regarding temperature. Please explain what is meant here, and support your claim.
Temperature matrices can be calculated similarly to SAR matrices, so that similar quadratic forms yield the temperature increase after a predefined heating time (under the assumption of constant perfusion and neglecting any other temperature-dependent phenomena). We have clarified this by adding an appropriate section in the methods (2.1).
Page 13 Line 430: Please define mean elevation factor before using it in results.
We have clarified this in the revised version of the manuscript.
Please report TC20, TC50 and TC80 values for the results given in Figure 6 and 7 to be in line with other hyperthermia studies and homogenization of the report of the study results.
We have added TC25, TC50 and TC 80 values to the result figures and also for the newly added comparison with the FOCO method.
Reporting an actual SAR distribution for a slice for head model (such as slice shown in Figure 3 d or e) would be beneficial. Maximum intensity projections smooth out the results and the behaviors at the tissue interfaces are not easily visible.
We have added additional non-projected slice plots to all figures in the revised version of the manuscript.
Please clarify in the figure captions that the reported SAR values above the projections are mean values in the target region.
We have clarified this in the captions.
Claims in the conclusion or in the abstract are weakly (or not) backed by quantitative results. Claims such as high performance, ideally suited for ideally suited for interactive hyperthermia treatment planning (what is the runtime?) are empty when the quantifications are not done. RF hyperthermia is already in use in the clinic as adjuvant as an adjunct anti-cancer treatment option. It is hard to see the merit for clinical use without quantitatively backed results.
We would like to thank the reviewer for the critical comments. To address the referee’s remark, the manuscript now includes a rigorous performance benchmark section to support the statements in the conclusion section and the abstract.
Reviewer 4 Report
This paper deals with the widely acknowledged complex optimisation problem in multi-antenna RF systems, particularly when applied to heating patterns in the strongly heterogeneous human body.
The authors have addressed this problem in an interesting and fundamental approach and show convincing examples of the benefits of its application in a variety of examples. I feel this manuscript is an excellent contribution to knowledge and will bring this field further ahead.
The discussion is somewhat brief but adequate, the applications in for instance RF safety and also the limitations are discussed.
Author Response
Thank you for your comments. Owing to the newly added performance benchmarks, the discussion section was also extended
Round 2
Reviewer 3 Report
Authors answered all my questions and clarified and added all my remarks from the previous review cycle in the manuscript. The merit of their study was much easier to appreciate with all the quantified and benchmarked results.